# Structural basis of Fumosorinone-mediated allosteric inhibition of PTP1B for cancer immunotherapy
Jun Zhang [1,2,3,9], Lianyun Lin[3,4,9], Nanxin Gong [1,5], Weikang Li[1], Yu Liu[1], Rajamanikandan Sundarraj[2,3], Yilong Li[2,3], Meijing Dong[6], Junan Ma[7], Kenneth Woycechowsky[8], Kunrong Mei[3], Yang Ge [3] ✉, Zhiguang Yuchi [2,3] ✉ & Duqiang Luo [1] ✉

Protein Tyrosine Phosphatase 1B (PTP1B) is a key immune regulator in cancer and an attractive immunotherapy target, yet progress is limited by the lack of selective inhibitors. Here, we identify Fumosorinone (FU), a natural product from *Isaria fumosorosea*, as a potent and selective allosteric inhibitor of PTP1B. In a murine colon tumor model, FU enhances anti-tumor immunity by reshaping the microenvironment, strengthening CD8⁺ T-cell responses, and promoting M1-like macrophage polarization. Enzymatic and biophysical analyses confirm its potency and direct engagement with PTP1B. A co-crystal structure defines a previously uncharacterized allosteric pocket that stabilizes the inactive state of the enzyme. This pocket is poorly conserved across the PTP family, consistent with minimal activity toward related phosphatases except TCPTP. Guided by this insight, virtual screening identifies additional inhibitors. These findings provide a structural basis for selective PTP1B targeting and support future immunotherapy development and rational drug discovery efforts.

Protein Tyrosine Phosphatase 1B (PTP1B) is a member of the protein tyrosine phosphatase (PTP) family, which comprises 107 members, including TCPTP, SHP1, and SHP2[1]. Members of this family function by dephosphorylating tyrosine residues on target proteins to regulate a wide array of cellular signaling pathways. Among them, PTP1B plays essential roles in various physiological contexts, particularly in the regulation of insulin and leptin signaling pathways, which are critical for maintaining glucose homeostasis and energy balance[2–5]. Beyond metabolism, PTP1B also influences immune signaling, modulating T-cell activation, cytokine signaling, and macrophage polarization. Dysregulation of PTP1B has been linked to several diseases, including cancer, obesity, and type 2 diabetes. In the context of cancer, PTP1B contributes to tumor progression by suppressing anti-tumor immune responses, enabling immune evasion, and reshaping the tumor immune microenvironment (TIME). Mechanistic studies have shown that PTP1B attenuates JAK/STAT signaling through the dephosphorylation and inactivation of JAK2 and TYK2, thereby impairing

T-cell expansion and activation, effectively functioning as an immune checkpoint[6–11]. These diverse physiological and pathological roles highlight PTP1B as a promising target for cancer immunotherapy.

Drug discovery efforts targeting PTP1B have focused on two main types of binding sites: the active site and various allosteric sites. Structurally, PTP1B consists of 435 amino acids organized into three regions: an N-terminal catalytic domain (residues 1–299), a central proline-rich regulatory domain (residues 300–400) that mediates protein-protein interactions, and a C-terminal tail (residues 401-435) that anchors the protein to the endoplasmic reticulum (ER) membrane[2,12]. The catalytic domain contains the conserved PTP signature motif $(HCX_5R)$, which is essential for enzymatic activity. This active site features a deep, positively charged pocket that accommodates negatively charged phosphotyrosine residues from substrates[13,14]. Although active-site inhibitors, such as benzylphosphonic acid derivatives, can effectively block catalytic activity by mimicking phosphotyrosine, their high polarity and limited selectivity, due to

[1]College of Life Sciences, Institute of Life Science and Green Development and Hebei Innovation Center for Bioengineering and Biotechnology, Hebei University, Baoding, China. [2]State Key Laboratory of Elemento-Organic Chemistry, College of Chemistry, Nankai University, Tianjin, China. [3]State Key Laboratory of Synthetic Biology and Frontiers Science Center for Synthetic Biology, School of Pharmaceutical Science and Technology, Faculty of Medicine, Tianjin University, Tianjin, China. [4]College of Bee Science and Biomedicine, Fujian Agriculture and Forestry University, Fuzhou, Fujian, China. [5]State Key Laboratory of Medicinal Chemical Biology, College of Life Sciences and College of Pharmacy, Nankai University, Tianjin, China. [6]Department of Bioinformatics, School of Basic Medical Sciences, Tianjin Medical University, Tianjin, China. [7]Department of Chemistry, State Key Laboratory of Synthetic Biology, Tianjin University, Tianjin, China. [8]School of Molecular Sciences, Arizona State University, Tempe, AZ, USA. [9]These authors contributed equally: Jun Zhang, Lianyun Lin. ✉e-mail: yangge@tju.edu.cn; yuchi@nankai.edu.cn; duqiangluo@hbu.edu.cn

conservation of catalytic residues among PTP family members, have posed significant challenges, particularly in achieving cell permeability and avoiding off-target effects[15–21].

To overcome these limitations, recent research has shifted toward targeting allosteric sites, regions outside the catalytic center that modulate enzyme function indirectly. The structural complexity and flexibility of PTP1B provide multiple opportunities for identifying such sites. To date, 12 allosteric pockets have been identified: 7 located within the catalytic domain and 5 within the regulatory domain[15,22–24]. These pockets offer enhanced druggability and selectivity by stabilizing inactive conformations or disrupting key regulatory interactions. A prominent example is Trodusquemine (MSI-1436), a natural spermine-cholesterol conjugate that selectively inhibits PTP1B by binding to its regulatory domain[24]. Its success led to the development of DPM-1003, a next-generation allosteric inhibitor currently undergoing Phase 1 clinical trials for Rett syndrome, announced in 2024.

In our recent work, we identified Fumosorinone (FU), a pyridinone alkaloid isolated from the fungus *Isaria fumosorosea*, as a potential PTP1B inhibitor[25]. However, the precise molecular mechanism of action and its potential in cancer immunotherapy remained unclear. In this study, we demonstrate that FU suppresses tumor growth in vivo by modulating the TIME. Specifically, FU enhances CD8$^+$ T cell activity and promotes M1-like macrophage polarization, while also inducing immunogenic cell death (ICD), thereby stimulating adaptive immune response. These findings highlight FU's potential as an immunotherapeutic agent. Biochemistry assays confirmed that FU inhibits PTP1B activity through a non-competitive mechanism, and structural analysis of the PTP1B-FU co-crystal complex revealed that FU binds to a previously uncharacterized allosteric pocket near the C-terminal region of the catalytic domain, locking the enzyme in an inactive conformation. Notably, bioinformatic analysis revealed that this allosteric pocket is poorly conserved across the PTP family, with significant similarity observed only in TCPTP, a closely related phosphatase that also plays a role in cancer and has been explored as a therapeutic target. Consistently, FU selectively inhibits PTP1B and TCPTP, while showing minimal activity against other PTPs, highlighting its narrow and therapeutically relevant subtype selectivity. Using this structural insight, we performed virtual screening of over one million compounds targeting the same site and identified several inhibitors with potent in vitro activity against PTP1B. Collectively, these findings identify an allosteric site on PTP1B and position FU as a promising lead compound for the development of selective PTP1B inhibitors, offering a strategy for cancer immunotherapy.

## Results

### FU potently inhibits tumor growth in mice

PTP1B plays a critical role in tumor immunity and has been identified as an effective immune checkpoint[7]. We investigated whether FU, as a PTP1B inhibitor, could suppress tumor progression by modulating antitumor immune responses (Fig. 1a). In MC38 tumor-bearing mice, FU treatment significantly inhibited tumor growth and extended median survival by 4days compared with the control group (Fig. 1b–d).

To evaluate the immunomodulatory effects of FU within the tumor microenvironment (TME), tumors were analyzed by multicolor flow cytometry. FU-treated tumors showed a marked increase in CD4$^+$T cells expressing γ-interferon (IFN-γ) and in CD8$^+$T cells expressing granzyme B (Gzmb) or IFN-γ, indicating enhanced cytotoxicity and effector functions of both T-cell subsets (Fig. 1e, f). Analysis of myeloid cells revealed no significant changes in overall myeloid cell populations, but FU altered tumor-associated macrophage (TAM) polarization: pro-inflammatory M1-like macrophages (iNOS$^+$) increased, whereas anti-inflammatory M2-like macrophages (CD206$^+$) decreased (Fig. 1g). These findings suggest that FU improves the TME not only by activating T cells but also by shifting the functional bias of TAMs toward an antitumor phenotype. We next assessed whether FU induces immunogenic cell death (ICD) in MC38 cells. FU treatment led to hallmarks of ICD, including increased calreticulin exposure, ATP release, and HMGB1 release (Fig. 1h, i).

Most immunotherapies are more effective against hot tumors, which are characterized by robust immune infiltration, than against cold tumors, which exhibit poor immune cell infiltration and immunosuppression[26,27]. Because MC38 is a hot tumor model, we also evaluated FU in a cold tumor model using MB49 bladder cancer cells, which are less responsive to immunotherapy. FU significantly inhibited tumor growth in the MB49 model, although the inhibition was approximately 1.3-fold weaker than in MC38 (Supplementary Fig. 1a, b). These results indicate that FU has potent antitumor activity in both hot and cold tumor models, supporting its potential for broad-spectrum antitumor effects.

Importantly, FU treatment showed no evident toxicity. Throughout the treatment period, there was no significant weight loss (Supplementary Fig. 1c, d) and no marked changes in serum ALT or AST levels (Supplementary Fig. 1e, f) compared with controls, suggesting that FU is well tolerated in vivo.

### Discovery of a non-competitive inhibitor of PTP1B

To elucidate the mechanism of FU inhibition, enzymatic kinetic assays were performed using p-nitrophenyl phosphate (pNPP) as the sole substrate. Recombinant human PTP1B (residues 1-299) was used in all assays. FU inhibited PTP1B with an IC$_{50}$ of 5.3 μM, in contrast to 225.3 μM for the sodium orthovanadate positive control (Fig. 2a). Michaelis–Menten analysis yielded a $K_m$ value of 2.4 mM. Notably, $V_{max}$ values decreased progressively with increasing concentrations of FU, while $K_m$ remained constant. Lineweaver-Burk plot (1/V vs. 1/[pNPP]) showed a family of lines intersecting near the x-axis in the second quadrant, consistent with non-competitive inhibition (Fig. 2b and Table 1). These findings suggest that FU inhibits PTP1B activity by decreasing catalytic efficiency ($V_{max}$) without affecting substrate affinity ($K_m$).

Direct binding between FU and PTP1B was further validated by isothermal titration calorimetry (ITC), yielding a dissociation constant ($K_d$) of 5.6 μM (Fig. 2c). Microscale thermophoresis (MST) showed a concentration-dependent decrease in normalized fluorescence, saturating at higher concentrations of FU, indicative of a specific and high-affinity interaction with a $K_d$ value of 6.8 μM (Fig. 2d).

### X-ray crystal structure reveals FU binding at a distinct allosteric site

PTP1B contains an N-terminal catalytic domain, a central proline-rich regulatory domain, and a C-terminal tail that anchors it to the ER membrane (Fig. 3a). Whereas the catalytic domain adopts a well-defined structure, the regulatory domain and C-terminal tail are largely intrinsically disordered. To elucidate how FU exerts its non-competitive inhibition, we determined high-resolution crystal structures of the apo PTP1B catalytic domain (hereafter referred to simply as PTP1B) and its FU-bound complex. FU binding does not alter the monomeric state of PTP1B in solution (Supplementary Fig. 2a), but it induces a change in crystal form and space group (Supplementary Fig. 2c and Table 2).

The co-crystal structure, resolved at 2.5 Å resolution, revealed that FU binds to an allosteric site formed between helices α3 and α6, but distal from the catalytic pocket (Fig. 3a, b and Supplementary Fig. 3). This binding site partially overlaps with the previously reported allosteric inhibitor site observed in the PTP1B-BBR complex (PDB1T49[22]), but extends toward adjacent residues that were not well characterized in earlier studies. Several key residues are involved in FU binding. The pyridine core and flexible aliphatic tail of FU fit snugly into a hydrophobic pocket formed by E276, K279, and F280, establishing stabilizing hydrophobic interactions. Additionally, the phenol group of FU engages in a hydrogen bond with N193 and makes a π-π stacking interaction with F280, while E276 forms a hydrogen bond with the N-hydroxy group of the pyridine ring, further stabilizing the ligand within the binding site (Supplementary Fig. 4a).

Mutagenesis confirmed the importance of specific residues in FU binding. Substitutions of residues on helix α3 that are in proximity to the phenol ring of FU, such as N193A, F196A, and F196R, led to reduced inhibition, with the N193A mutation showing the most significant decrease

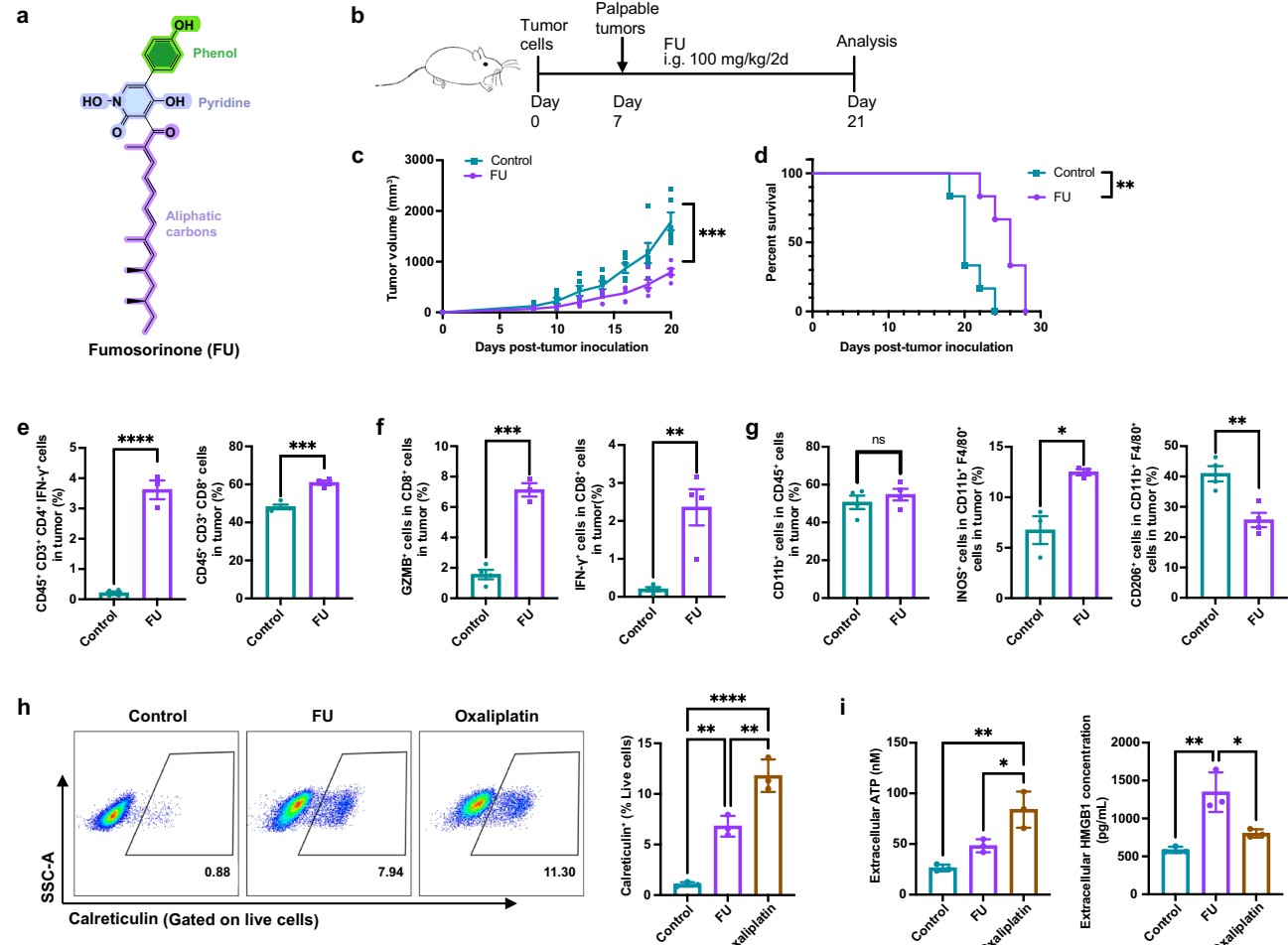

**Fig. 1 | FU inhibited tumor growth in vivo by modulating the TIME. a** Chemical structure of FU. **b** Experimental scheme for FU treatment in the MC38 tumor-bearing mouse model. Mice were subcutaneously inoculated with MC38 cells on day 0 and treated with FU or PBS (control) every two days from day 7 to day 19. **c** Tumor volume curves over time ($n = 6$). FU significantly reduced tumor growth. **d** Kaplan–Meier survival analysis of MC38-bearing mice ($n = 6$); significance assessed by log-rank test. **e–g** Flow cytometry analysis of tumor-infiltrating immune cells on day 13. **e** FU increased the proportion of IFN-γ⁺ CD4⁺ T cells (left) and CD8⁺ T cells (right) among CD45⁺ cells. **f** Among CD8⁺ T cells, FU elevated GZMB⁺ (left)

and IFN-γ⁺ (right) populations. (**g**) While total CD11b⁺ myeloid cell levels remained unchanged (left), FU increased pro-inflammatory M1-like macrophages (iNOS⁺) and decreased anti-inflammatory M2-like macrophages (CD206⁺) among TAMs (CD11b⁺ F4/80⁺). **h, i** Assessment of immunogenic cell death (ICD) in MC38 cells. (**h**) FU or oxaliplatin increased surface calreticulin levels (flow cytometry). **i** Extracellular ATP and HMGB1 release were elevated after FU or oxaliplatin treatment (luminescence assays). Data are presented as mean ± SEM. Statistical significance determined by an unpaired *t*-test. ns = not significant; *$p < 0.05$; **$p < 0.01$; ***$p < 0.001$; ****$p < 0.0001$.

in inhibition. Consistent with these functional effects, ITC measurements performed under identical conditions showed no detectable FU binding to these mutants, indicating that their residual affinities were weakened beyond the reliable detection limit of this method (Supplementary Fig. 5). By contrast, the E276A, located at helix α6 within the hydrophobic pocket, abrogated enzyme activity entirely, precluding evaluation of inhibition. Mutations in nearby helix α7, including W291A, had no significant effect on FU inhibition, indicating the interaction is primarily mediated by residues in helix α3 and adjacent hydrophobic regions (Fig. 3c).

## FU prevents WPD loop closure and traps PTP1B in an inactive conformation

PTP1B recognizes its substrates through the dynamic WPD loop (residues 179-187) and catalyzes the dephosphorylation of phospho-tyrosine (pTyr) to tyrosine (Tyr) via the catalytic C215 residue located within the PTP loop (residues 213-223). Structural studies have shown that PTP1B adopts both open and closed conformations, primarily regulated by the movement of the WPD loop[28]. In the open conformation, the WPD loop swings outward, creating an accessible active site for substrate entry. In the closed

conformation, typically stabilized by substrate binding, the WPD loop folds over the catalytic pocket to facilitate catalysis.

We used the PTP1B-substrate complex (PDB ID: 1G1G) as a representative of the closed conformation[29], and the apo structure of unbound PTP1B as the reference for the open state. Comparison of PTP1B-FU with the apo structure revealed minimal structural deviation (RMSD = 0.45 Å over 275 pruned atom pairs), suggesting that FU binding stabilizes a conformation similar to the open, catalytically inactive state (Supplementary Fig. 6). Alignment of the Cα backbone atoms in the WPD loop, as well as helices α3 and α6, further confirmed this resemblance.

In contrast, when compared with the substrate-bound closed conformation, the FU-bound structure shows significant shifts in the positions of helices α3 and α6 and in the backbone trajectory of the WPD loop, particularly where the loop interacts with α3. Given the essential role of WPD loop closure in catalysis, these findings suggest that FU prevents loop closure by stabilizing an open-like conformation, thereby blocking enzymatic activity.

Structural analysis of the PTP1B-FU complex revealed that FU forms specific interactions with residues on helix α3 (P188, L192, N193, F196) and

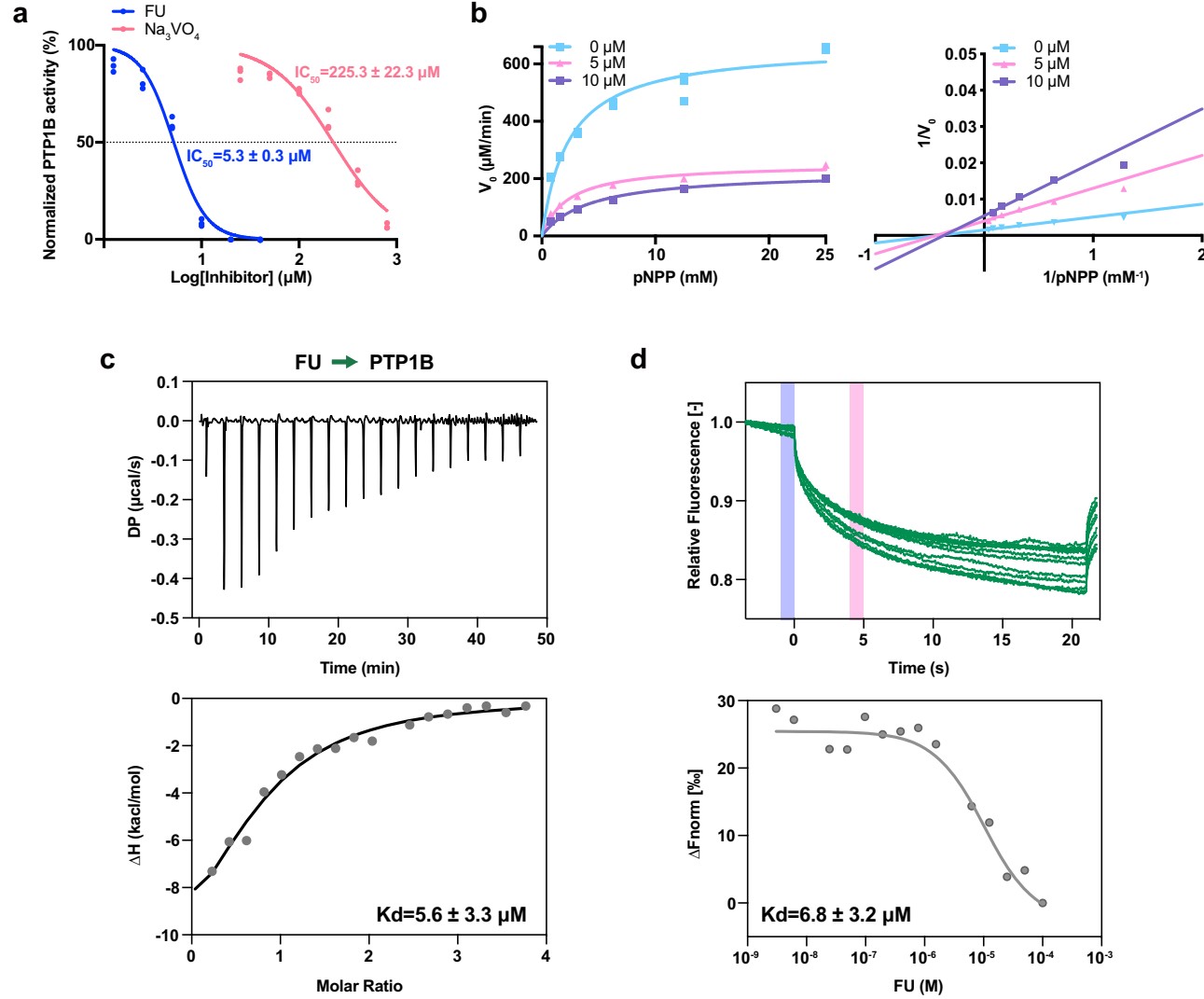

**Fig. 2 | FU functions as a non-competitive inhibitor of PTP1B. a** Dose–response curve showing inhibition of PTP1B by FU; sodium orthovanadate (Na$_3$VO$_4$) was used as a positive control. **b** Michaelis–Menten (left) and Lineweaver-Burk (right) plots of PTP1B activity with varying FU concentrations (0.5, 10 μM) and varying substrate concentrations (0–25 mM). **c** ITC showing FU binding to PTP1B with a $K_d$ of 5.6 μM. **d** MST analysis confirming that FU binds directly to PTP1B with a dissociation constant ($K_d$) of 6.8 μM.

## Table 1 | Enzyme activity constants for PTP1B

| FU (μM) | $V_{max}$ (μM/min)[a] | $K_m$ (mM)[b] | $K_{cat}$ (s$^{-1}$)[c] | $K_{cat}/K_m$ (s$^{-1}$/mM) |
|---|---|---|---|---|
| 0 | 664.8 ± 20.4 | 2.4 ± 0.1 | 22.2 ± 0.7 | 9.5 ± 0.1 |
| 5 | 251.8 ± 10.4 | 2.3 ± 0.1 | 8.4 ± 0.3 | 3.7 ± 0.1 |
| 10 | 224.4 ± 4.6 | 4.2 ± 0.5 | 7.5 ± 0.2 | 1.8 ± 0.2 |

[a]$V_{max}$: maximum reaction velocity.

[b]$K_m$: Michaelis–Menten constant representing the substrate concentration at half $V_{max}$.

[c]$K_{cat}$: catalytic turnover number (number of substrate molecules converted per enzyme molecule per second).

helix α6 (E276, K279, F280), effectively locking these helices in place (Fig. 3d). This conformational constraint appears to hinder the WPD loop from transitioning into its closed state, preventing proper substrate positioning and catalytic turnover, a key mechanism underlying the non-competitive inhibition by FU (Fig. 3e).

## FU selectively modulates PTP1B and TCPTP

Members of the PTP family share high sequence and structural similarity, making the development of selective inhibitors for individual PTP members

particularly challenging. Structural analysis of the PTP1B-FU complex revealed that FU binds an allosteric site located between helices α3 and α6 within the catalytic domain. To assess the conservation of this site across the PTP family, we performed sequence alignment and structural comparison. Phylogenetic analysis showed that only a small subset of PTP family members possesses conserved FU-contacting residues from both helices α3 and α6, with PTP1B and TCPTP being the only members sharing this structural feature (Fig. 4a, b). TCPTP, which shares >70% sequence identity with PTP1B, functions as a negative regulator of immune signaling by dephosphorylating JAK and STAT proteins that control T cell and macrophage differentiation[30–34]. Loss of TCPTP can lead to immune dysregulation[20]. Prior studies have demonstrated that dual targeting of PTP1B and TCPTP exerts anti-tumor effects by directly modulating tumor cell behavior and enhancing immune cell function[6].

To evaluate FU's selectivity, we conducted in vitro enzyme kinetic assays on a series of closely related PTP family members, including TCPTP, LYP, STEP and RPTPα. FU inhibited TCPTP with an IC$_{50}$ of was 12.6 μM (Fig. 4c), whereas it showed no significant inhibition of the other phosphatases, indicating a high degree of selectivity for PTP1B and its closest homolog TCPTP. We also conducted homology modeling of the TCPTP and FU complex structure using the PTP1B-FU complex as a template. The

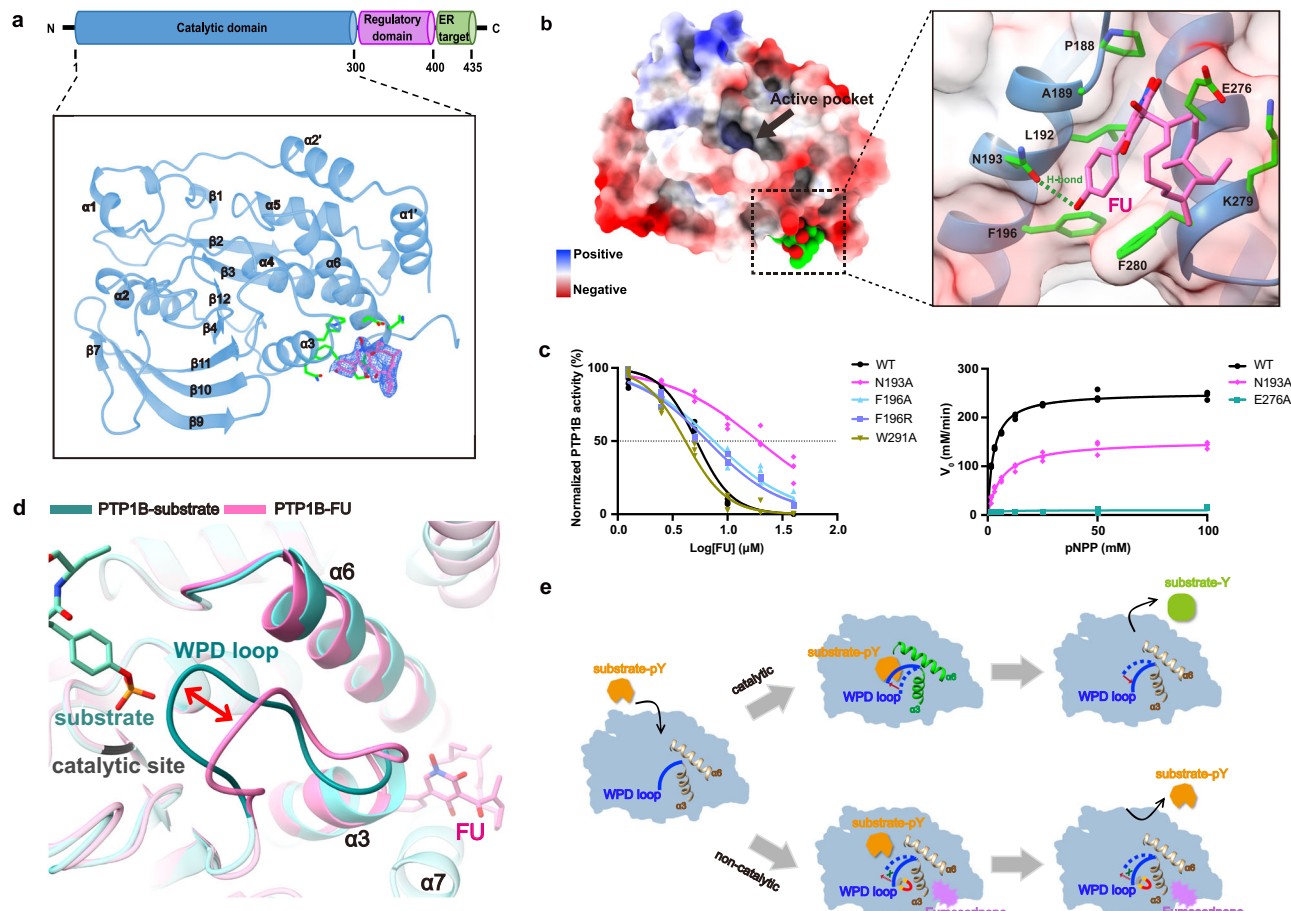

**Fig. 3 | Crystal structure of the PTP1B-FU complex and identification of key binding residues. a** Overall structure of the complex. FU is shown in pink, key residues in green, and the 3.4σ electron density map as a blue mesh. **b** Electrostatic surface potential of the complex (blue = positive, red = negative). FU binds to Pro188, Ala189, Leu192, Asn193, Phe196, Glu276, Lys279, and Phe280, stabilizing helices α3 and α6. **c** Inhibitory effects of FU on WT PTP1B and key mutants (N193A, F196A, F196R, W291A); enzyme activity assays for N193A and E276A confirm suppressed activity. **d** Superimposition of FU-bound PTP1B (hot pink) with substrate-bound structure (turquoise, PDB ID 1G1G) shows that FU prevents WPD loop closure. **e** Proposed model of FU-mediated inhibition of PTP1B.

results indicated a high degree of similarity between them (Supplementary Fig. 7). Of the seven residues forming the FU-binding pocket, six are identical, with the only difference being Phe280 in PTP1B replaced by a cysteine in TCPTP. This substitution likely accounts for FU's ~2-fold higher inhibitory potency against PTP1B.

### Identification of a FU-site-targeting inhibitor

Most PTP1B inhibitors designed to target the catalytic pocket are limited by poor cell permeability and lack of PTP isoform selectivity, owing to the positively charged nature and high structural conservation of the catalytic site. In contrast, the FU binding site is located in an allosteric region distant from the catalytic center. This region is less conserved among PTP family members, making it an attractive site for the development of PTP1B-selective inhibitors with greater specificity and reduced off-target effects.

To discover inhibitors targeting the FU site, we performed virtual screening using our high-resolution FU-PTP1B complex structure as a template (Fig. 5a). From a library of 1,270,000 compounds, the top 107 candidates were selected based on docking scores for subsequent enzymatic validation (Fig. 5b). Of these, 17 compounds with chemical scaffolds distinct from FU exhibited significant inhibitory activity against PTP1B at a concentration of 20 μg/mL (Fig. 5c). Notably, one compound, designated PI-1, showed an IC$_{50}$ of 4.7 μM, comparable to that of FU (Fig. 5d). Docking and molecular dynamics (MD) simulation analysis revealed that PI-1 interacts with N193, F196, and F280 within the FU binding site, with a docking score of -7.269 kcal/mol, suggesting a similar inhibitory mechanism (Fig. 5e, f and Table 3). In the final MD-stabilized conformation, the aromatic rings of PI-1

consistently formed π–π stacking interactions with F196 and F280, while N193 maintained a stable hydrogen bond network with the ligand, collectively stabilizing the complex (Supplementary Fig. 4b).

### Discussion

In recent years, PTP1B has gained considerable attention as an important immunotherapeutic target. In this study, we demonstrate that FU, a PTP1B inhibitor, exhibits significant antitumor activity in both colon and bladder cancer mouse models, primarily by modulating the TIME. FU markedly enhances the infiltration and activation of CD8$^+$ T cells and M1-like macrophages, reshaping the TIME toward an antitumor state. Notably, FU also induces ICD in tumor cells, further amplifying antitumor immunity and highlighting FU's potential as a lead compound for broad-spectrum antitumor drug development.

Although PTP1B is well established as a tumor-associated target, efforts to develop PTP1B inhibitors have been hampered by challenges in achieving sufficient selectivity and cell permeability, largely due to the highly conserved and positively charged nature of its catalytic pocket. To date, only four crystal structures featuring a single class of compounds bound to a distinct allosteric site of PTP1B have been reported[22,35]. Our co-crystal structure of the PTP1B-FU complex reveals a previously uncharacterized allosteric site that provides a promising framework for the design of PTP1B-selective inhibitors. FU binding stabilizes the WPD loop in an open conformation, thereby preventing the closure of the catalytic pocket that is essential for substrate turnover. This mechanism suggests that other small molecules capable of locking the WPD loop in an open state may also exert

**Table 2 | Data collection and refinement statistics for crystal structures**

| Crystal | PTP1B-APO | PTP1B-FU |
|---|---|---|
| λ for data collection (Å) | 0.9414 | 0.9791 |
| Data collection | | |
| Space group | P 31 | P 43 |
| *Cell dimension* (Å) | | |
| *a, b, c* (Å) | 86.457, 86.457, 103.345 | 87.962, 87.962, 165.075 |
| *α, β, γ* (°) | 90, 90, 120 | 90, 90, 90 |
| Resolution | 33.16–2.407 | 39.34–2.5 |
| Rmerge | 0.164 (0.937) | 0.163 (0.738) |
| Average $I/σ(I)$ | 8.9 (1.6) | 8.9 (2) |
| Completeness (%) | 98.91 (92.22) | 99.88 (99.91) |
| Redundancy | 6.8 (6.0) | 6.8 (6.9) |
| Refinement | | |
| Resolution (Å) | 33.16–2.407 | 39.34–2.5 |
| Highest resolution shells (Å) | 2.493–2.407 | 2.59–2.5 |
| No. of reflections | 55,499 | 43,292 |
| Average *B*-factor | 41.06 | 32.51 |
| *Rwork* | 0.1773 (0.2526) | 0.1797 (0.2018) |
| *Rfree* | 0.2126 (0.2913) | 0.2230 (0.2616) |
| RMSD length (Å) | 0.008 | 0.007 |
| RMSD angle (°) | 0.98 | 0.80 |
| No. of atoms | | |
| Protein | 4524 | 4853 |
| Ligands | 24 | 94 |
| Water | 129 | 248 |
| Ramachandran plot (%) | | |
| Most favored | 93.71 | 94.76 |
| Additionally allowed | 3.78 | 3.14 |

Values in parentheses refer to the highest resolution shell.

potent inhibitory effects. In addition to the FU site, the neighboring region formed by helices α3 and α7 and strand β10 may represent another potential allosteric site capable of modulating WPD loop dynamics and inhibitor binding.

The functional importance of the FU site is further supported by mutational analyses. Mutations of key residues within this pocket impaired enzymatic activity without affecting PTP1B expression. In particular, the E276A mutation led to a near-complete loss of catalytic activity, while N193A markedly reduced activity. These residues, together with K197 and helix α7, are positioned within the structural coupling network surrounding the WPD loop, which governs the conformational transition required for catalysis. Thus, the detrimental effects of E276A and N193A are mechanistically consistent with previous reports showing that K197A substitution or deletion of helix α7 significantly reduces PTP1B activity[35]. Together, these data underscore the functional relevance of the FU site and validate it as a promising target for the design of modulatory inhibitors.

FU displayed strong selectivity for PTP1B and TCPTP over other PTP family members. This dual targeting could be therapeutically advantageous, as both PTP1B and TCPTP contribute to tumor cell proliferation and immune regulation[6,7,32,36]. Given their complementary roles in key signaling pathways, simultaneous inhibition of PTP1B and TCPTP may achieve more comprehensive pathway blockade and potentiate antitumor effects.

While our biochemical, structural, and in vivo data strongly support the inhibitory role of FU on PTP1B, additional cellular-level validation will further strengthen the mechanistic framework. In particular, future studies examining the phosphorylation status of canonical PTP1B substrates and downstream signaling nodes in tumor-infiltrating CD8$^+$ T cells will be important for directly linking FU treatment to intracellular immune signaling rewiring. Such experiments will provide complementary evidence for the cellular mechanism of FU and help further define how PTP1B inhibition reshapes antitumor immunity.

In summary, our work provides a detailed mechanistic characterization of FU, a natural small-molecule PTP1B inhibitor, through an integrated structural and functional approach. We further demonstrate its robust efficacy in remodeling tumor immunity and suppressing tumor growth in vivo, thereby expanding the therapeutic potential of PTP1B as an immune checkpoint target. Beyond establishing a structural basis for allosteric inhibitor design, these findings also provide a generalizable screening and optimization strategy for developing next-generation selective PTP1B modulators. Future head-to-head efficacy and immune-profiling comparisons with benchmark inhibitors such as AC484 will further clarify the translational positioning of this chemotype within the emerging landscape of PTP1B-targeted immunotherapeutics. Importantly, the structural insights revealed here offer a clear roadmap for the medicinal chemistry optimization of FU and related scaffolds, including PI-1, to further improve pharmacokinetic properties, in vivo stability, and target selectivity, ultimately accelerating the development of more precise and effective therapies.

## Materials and methods
### Mice and cell lines
Female C57BL/6 J mice (6–8 weeks, approximately 20 g) were purchased from Shanghai Model Organisms Center, Inc. and housed under specific pathogen-free (SPF) conditions at the Nankai University Animal Resources Center. All animal care and experimental procedures were approved by the Animal Ethics Committee of Nankai University (Approval No. 2023-SYDWLL-000297). MC38 and MB49 cell lines were purchased from the American Type Culture Collection (ATCC, Virginia, USA) and are stored in the laboratory. Cells cultured in DMEM supplemented with 10% fetal bovine serum (FBS, 164210, Procell) and 1% Penicillin/streptomycin (P/S, PB180120, Procell). FU was isolated and purified by the research group of Prof. Duqiang Luo.

### Animal experiment
C57BL/6 J mice ($n = 6$ per group) were subcutaneously implanted with $1 \times 10^6$ MC38 or MB49 cells. When tumors reached approximately 50–100 mm$^3$ on day 7, FU and PBS control were administered orally (100 mg/kg, every 2 days). The number of mice was decided according to previous reports with preliminary data on tumor growth variance. Tumor volume and body weight were monitored every two days. All animal experiments were conducted in accordance with the guidelines of the Nankai University Research Institute Research Ethics Committee (protocol registration number: A-2018-0306). At the end of the treatment period, mice were euthanized by gradual CO₂ inhalation in accordance with ethical guidelines to minimize pain and distress. During the evaluation of FU's antitumor efficacy in colon cancer, mouse body weight was recorded throughout treatment, and serum alanine aminotransferase (ALT) and aspartate aminotransferase (AST) levels were measured at the end of the study to assess liver function. After the treatment was completed, tumor volumes were measured to evaluate antitumor efficacy. The excised tumors were subsequently digested into single-cell suspensions for flow cytometry analysis.

### Flow cytometry
After tumor-bearing mice were treated with FU three times, tumor tissues were isolated, and flow cytometry was used to analyze the infiltration ratio

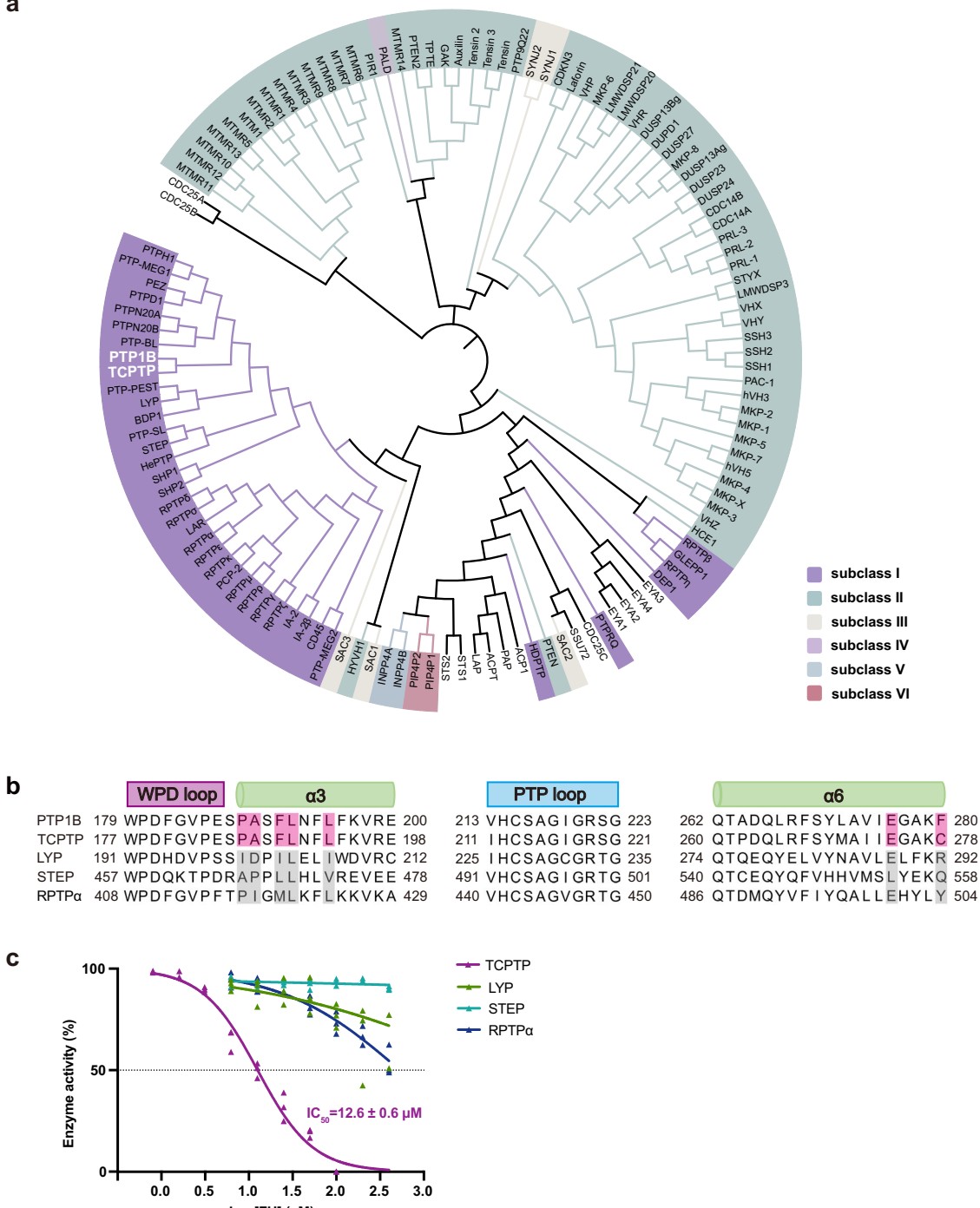

**Fig. 4 | Conservation analysis of the FU allosteric binding site across PTP family members. a** Phylogenetic tree illustrating the evolutionary relationships among six PTP subclasses within the class I PTP family. Only PTP1B and TCPTP, both belonging to subclass I, retain conserved residues on helices α3/α6 that constitute the FU allosteric binding site. **b** Sequence alignment of helices α3/α6, the WPD loop (residues 179–187) and the PTP loop (residues 213–223) among PTP1B and closely related subclass I members (TCPTP, LYP, STEP and RPTPα). Conserved residues shared by α3/α6 of PTP1B and TCPTP are highlighted in pink; non-conserved residues in other members are shown in gray. **c** FU also inhibits TCPTP, with an $IC_{50}$ of 14.7 μM.

and functional changes of tumor-infiltrating immune cells. Tumors were digested into single-cell suspensions and analyzed for changes in the infiltration rate and function of tumor-infiltrating T cells using flow cytometry. Flow cytometry was used to analyze the changes in the function of infiltrating macrophages in the tumor microenvironment. Tumor tissues were enzymatically dissociated into single-cell suspensions using a digestion solution containing Collagenase IV (1 mg/mL), Hyaluronidase (1 mg/mL), and DNase I (20 U/mL) (Sigma). The isolated cells were stained with a viability dye at room temperature for 15 min, followed by incubation with mouse anti-CD16/CD32 (156603) antibodies for 15 min. Subsequently, fluorescence-conjugated antibodies were applied for 30 min to label cell-surface markers (1:100 dilution), including CD45 (103108, 103126), CD3 (100218), CD4 (100428), CD8 (100752), CD25 (102036), CD69 (104512), NKp46 (137618), CD19 (152406), CD11b (101224, 101263,), Ly6C (128033), Ly6G (127639), F4/80 (123114), CD206 (141732), iNOS (696806), Gr-1 (108428), CD11c (117334), MHC-II (107614), PD-1

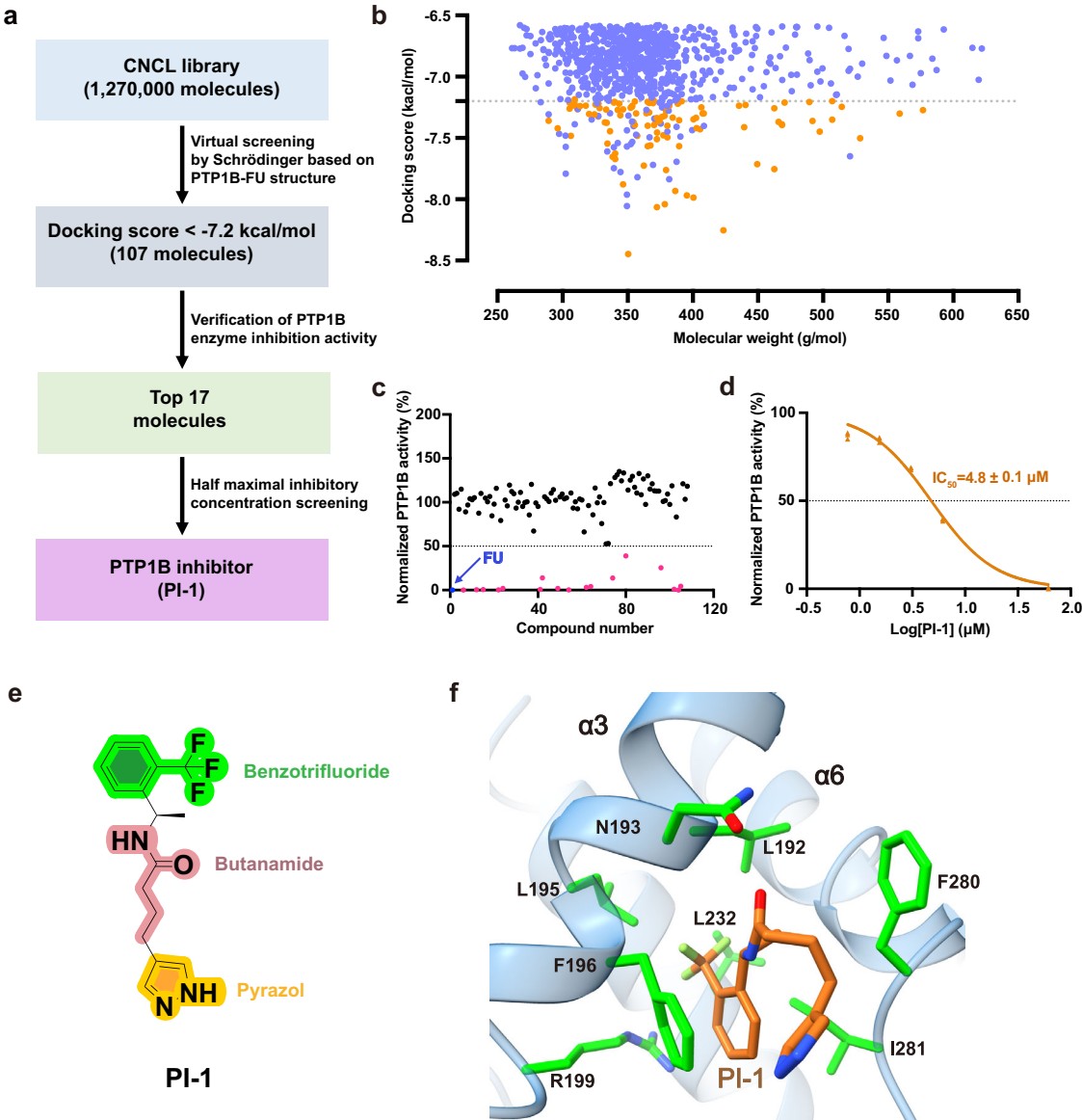

**Fig. 5 | Structure-based virtual screening of PTP1B inhibitors targeting the FU allosteric site. a** Workflow of virtual screening based on the PTP1B-FU complex structure. **b** Distribution of docking scores for screened compounds; top 107 hits are shown in orange. **c** In vitro inhibition of PTP1B by 107 compounds. The top 17 are in pink, FU in blue, and others in black. **d** Dose-response curve for lead compound PI-1. **e** Chemical structure of PI-1. **f** Predicted binding pose of PI-1 within the FU allosteric site; key interacting residues in green, PI-1 in brown.

## Table 3 | Docking result of the identified PTP1B inhibitor

| Compound ID | Docking score (kcal/mol) | Prime MM GBSA (kcal/mol) | H-bond interactions | π–π stacking |
|---|---|---|---|---|
| PI-1 | −7.269 | −51.662 | Asn193 | Phe196, Phe280 |

(135228) (all from BioLegend). After washing with PBS, cells were fixed, permeabilized, and subjected to intracellular staining for 30 min with GZMB (372208) and Foxp3 (126403) (BioLegend). Flow cytometric analysis was then performed. The gating strategy is shown in Supplementary Figs. 9 and 10.

### ICD evaluation
MC38 cells were inoculated into 6-well culture plates and cultured until the cell fusion was about 70-80%. Cells were treated with FU, oxaliplatin (MedChemexpress) was used as a positive control, and PBS as a blank control. After 24 h, the cells were collected to detect the characteristic markers of ICD. The ATP concentration in the culture supernatant was

determined by using an ATP assay kit (Sigma). HMGB1 concentration in culture supernatants is measured by ELISA using the HMGB1 Assay Kit (Sigma). Calreticulin assay: Cells were surface-stained with anti-CRT antibody (Sigma), then dispersed into single-cell suspensions through a flow cytometric sieve, transferred to a flow tube, and the fluorescence intensity of the cells was measured by a BD flow analyzer. The data were analyzed by FlowJo software. Three sets of replicates were set up for all experiments to ensure the reliability of the results.

### Expression and purification of recombinant human PTP1B
The cDNA encoding the PTP domain of human PTP1B (residues 1–299) was cloned into the pEASY-E1 plasmid and transformed into *E. coli* BL21.

Cells were grown in LB medium containing 1 mg/mL ampicillin at 37 °C with shaking until reaching an $OD_{600}$ of 0.6–0.8, and subsequently induced with 0.4 mM IPTG for overnight expression at 18 °C. After centrifugation at $8000 \times g$ for 10 min, pellets were resuspended in lysis buffer (10 mM HEPES, PH 7.4, 250 mM KCl, 25 mg/mL DNase I, 25 mg/mL lysozyme and 1 mM PMSF). Following centrifugation at $40,000 \times g$ for 30 min to remove cell debris, the supernatant was filtered through a 0.22 µm filter before loading onto a 5 mL His Trap HP column (GE Healthcare) equilibrated with buffer A (10 mM HEPES, pH 7.4, 250 mM KCl). The column was eluted with a linear imidazole gradient (20-250 mM) in buffer A. The eluted PTP1B sample was further purified using a Q Sepharose high-performance column (GE Healthcare) with a linear gradient from 20 to 500 mM KCl in elution buffer (10 mM Tris, pH 8.8). The purified PTP1B was concentrated using Amicon concentrators (10 K MWCO, Millipore) and subjected to a Superdex 75 pg 16/600 gel filtration column (GE Healthcare) in buffer A. Protein purity was assessed by 15% (wt/vol) sodium dodecyl sulfate polyacrylamide gel electrophoresis (SDS-PAGE) (Supplementary Fig. 2b). The final sample was concentrated to 10 mg/mL (10 mMHEPES, pH7.4, 50 mMKCl) and stored at −80 °C.

During purification of the PTP1B-FU mixture to assess whether FU binding alters the monomeric state of PTP1B, 10 µM FU was added at the initial cell lysis stage and maintained at each subsequent purification step to ensure sufficient ligand concentration for binding. The point mutations N193A, F196A, F196R, E276A, and W291A in PTP1B were generated by PCR (primer sequences provided in Supplementary Table 1), and the purification methodology is consistent with WT. Expanded culture for TCPTP, LYP, RPTPα and STEP is the same as PTP1B. The protein purity was examined by 12% (wt/vol) SDS-PAGE (Supplementary Fig. 8a–c).

### Determination of phosphatase activity of PTPs using pNPP substrate

To evaluate the phosphatase activity of PTP1B, p-nitrophenyl phosphate (pNPP) was used as the isolated substrate. The reaction mixture (100 µL total volume) contained 10 mM sodium acetate buffer (pH 5.5), 1 mM ethylenediaminetetraacetic acid (EDTA), 1 mM DL-dithiothreitol (DTT), and 2% dimethyl sulfoxide (DMSO). Recombinant PTP1B protein was incubated with varying concentrations of FU at 37 °C for 15 min. Enzyme activity was assessed immediately after the addition of pNPP by monitoring the formation of p-nitrophenol, measured continuously at 405 nm for 10 min at 37 °C using an Infinite 200Pro Microplate Reader (Tecan Trading AG). The final enzyme concentration after dilution was 0.5 µM. FU was tested against PTP1B at concentrations of 1.25, 2.5, 5, 10, 20, and 40 µM.

The kinetic assays for TCPTP, LYP, RPTPα, and STEP were conducted under identical conditions as for PTP1B. In the TCPTP assay, FU was initially prepared at 100 µM and serially diluted two-fold across eight concentrations. In the assays for LYP, RPTPα, and STEP, the starting FU concentration was 400 µM, followed by six two-fold serial dilutions.

### Isothermal titration calorimetry experiments

The purified WT and mutant PTP1B proteins were dialyzed overnight at 4 °C against a buffer containing 150 mM KCl and 10 mM HEPES (pH 7.4). FU was diluted in the same dialysis buffer. For titrations containing 4% DMSO, twenty injections of 2 µL FU solution were added to the sample cell containing 10 µM WT or mutant PTP1B protein, or buffer control. The reference cell was filled with water. All experiments were performed at a constant temperature of 25 °C with continuous stirring at 750 rpm using a PEAQ-ITC instrument (Malvern). Subsequently, the collected data were processed on GraphPad Prism version 8.0.0 (www.graphpad.com; San Diego, California, USA).

### Microscale thermophoresis experiments

MST measurements of FU binding to PTP1B were conducted using a Monolith NT.115 instrument (NanoTemper). PTP1B was fluorescently labeled with the Monolith™ RED-NHS Second Generation Protein Labeling Kit in buffer containing 150 mM KCl and 10 mM HEPES, pH 7.4.

For binding assays, the labeled PTP1B was diluted 100-fold in buffer supplemented with 0.1% Tween-20 to achieve optimal fluorescence intensity. FU was serially diluted by buffer (1:1 ratio) across 16 concentrations. Subsequently, equal volumes (10 µL each) of PTP1B and FU solutions were mixed and incubated for 5 min at room temperature before being loaded into standard glass capillaries. MST measurement was performed at 70–100% excitation power, maintaining fluorescence between 300 and 400 nm, using the MO control software (v.1.6.1). The thermophoresis time was 23 s, and the experimental temperature was 25 °C. Data analysis was performed using MO affinity analysis software (v.2.3, NanoTemper), applying the $K_d$ model. Cold and hot fluorescence signals were recorded at −1–0s and 4–5 s, respectively, to minimize thermally induced conformational changes in the protein.

### Crystallization, data collection, and structure determination

FU was prepared with 100% DMSO. The PTP1B and FU complex was prepared by mixing PTP1B (10 mg/mL) and FU at a 1:10 molar ratio, in 10 mM HEPES (PH 7.4), 50 mM KCl at 4 °C. Initial crystallization screening was performed by sitting-drop vapor diffusion using commercial sparse matrix screens (Hampton Research and Molecular Dimensions). Crystallization trials were set up in 96-well format at a 1:1 ratio with an automated liquid handling robotic system (Gryphon, Art Robbins). After initial hits, optimization was carried out by hanging-drop vapor diffusion in 24-well plates. Optimal crystals were obtained in 0.1 M MES, pH 6.0, 0.1 M calcium acetate hydrate, and 15% PEG 400. Crystals were mounted in Cryo-loops (Hampton Research) and flash-cooled in liquid nitrogen with reservoir solution supplemented with 20% glycerol as a cryoprotectant. Diffraction data were collected at beamline BL18U1 of the Shanghai Synchrotron Radiation Facility (SSRF) to resolutions of 2.5 Å (PTP1B and FU) and 2.4 Å (PTP1B). The final DMSO concentration in the PTP1B-FU complex solution was 5%. Data were indexed, integrated, and scaled using the HKL2000 suite[37]. Molecular replacement was performed in PHENIX[38] using the crystal structure of PTP1B in the apo state as a search model (PDB ID: 1SUG). The model sequence was replaced with the target sequence after Phaser-MR, and the structure was manually rebuilt in Coot[39] and iteratively refined in PHENIX. Data collection and refinement statistics are summarized in Table S1. All structure figures were generated with UCSF ChimeraX[40]. The 2D interaction diagram of the complex structure is generated by the website (https://proteins.plus).

### Bioinformatics analysis

The sequences of the FU allosteric site were extracted as templates. The sequence of helix α3 corresponds to motif 1 (PASFLNFLFKVRE) and the sequence of helix α6 corresponds to motif 2 (QTADQLRFSYLAVIEGAKF). The protein sequences of the PTP family were downloaded from the Universal Protein (Uniprot) database in FASTA format, and the specific classification details were referred to the reference[41]. Subsequently, all downloaded sequences were subjected to multiple sequence alignment using MEGA12 software. The sequences corresponding to motif 1 and motif 2 in each sequence were extracted through this process. Based on these extracted sequences, a phylogenetic tree based on the structural domain sequences was constructed using the Maximum Likelihood method to better understand the evolutionary relationship and conservation of this site in the PTP family.

Sequence identity of the PTP domain in PTP1B and other PTPs was estimated using Clustal Omega (https://www.ebi.ac.uk/Tools/msa/clustalo/). Multiple sequence alignment of the FU-site in non-receptor PTPs was performed using CLC Main Workbench 7.9.1. The PTP domains were selected based on the previous crystal structures.

### Homology modeling

The TCPTP sequence (GenBank ID: NM_002828.4) was retrieved from GenBank. We selected the complex structure of PTP1B and FU, which share 86% sequence identity with the catalytic domain of TCPTP, as the templates for homology modeling to create the TCPTP-FU complex structural model.

The structure was constructed using the Homology Modeling module of Schrödinger software[42]. The default parameters were used.

## Virtual screening on FU binding allosteric pocket

A total of 1,270,000 ligands from the Chinese National Compound Library (CNCL) were prepared using the LigPrep module in the Schrödinger Suite 2017[43]. The 2D ligand structures were converted into energy-minimized 3D structures using the OPLS3 force field, with additional conformations generated through a rapid torsion search in MacroModel[43,44]. Ligand preparation was performed at pH 7.0 ± 2.0, and tautomeric and ionization states were generated using the Epik module[43]. The protein structure was processed in Maestro (version 11.1)[43] using the Protein Preparation Wizard and energy-minimized with the OPLS3 force field. A receptor grid centered on Phe280 of helix α6 was generated using the Receptor Grid Generation module, and ligands were docked within a 20 Å radius of the centroid.

Virtual screening was conducted to identify selective inhibitors targeting PTP1B. Prior to the screening, the CNCL library was filtered through Lipinski's rule of five. The Glide docking program (version 7.4) was employed for virtual screening, which was carried out in a hierarchical manner: High-Throughput Virtual Screening (HTVS), followed by Standard Precision (SP), and finally Extra Precision (XP) docking[45]. Molecules exhibiting the highest Glide score and Glide energy were visually inspected and prioritized for further analysis.

## Molecular dynamics simulation

MD simulation was conducted using the Desmond 2021-4 module to evaluate the stability of the protein-ligand complexes[46]. An orthorhombic simulation cell was created and solvated with the transferable intermolecular potential3 points (TIP3P) water model. The OPLS-2005 force field was applied to optimize all atoms within the system, which was then neutralized by adding an appropriate number of $Na^+$ or $Cl^-$ counter ions at pH 7.4. Temperature (298 K) and pressure (1 atm) were maintained using the Nose-Hoover thermostat and Martyna-Tobias-Klein barostat, respectively[47]. Energy minimization was performed via the Steepest Descent algorithm to eliminate unfavorable van der Waals interactions, and long-range electrostatic interactions were computed using the particle mesh Ewald method. The production phase comprised a 100 ns MD simulation, with snapshots recorded every 100 ps. Root mean square deviation (RMSD) and root mean square fluctuation (RMSF) analyses were generated using the Simulation Interaction Diagram tool. Upon PI-1 binding to PTP1B, RMSD showed initial fluctuations during the first 70 ns, after which it stabilized, indicating the formation of a stable protein-ligand complex for the remainder of the simulation.

## Statistics and reproducibility

All experiments were performed with at least three independent biological replicates unless otherwise stated. Data are presented as mean ± SEM. Statistical analyses were carried out using GraphPad Prism 9.0. Differences between the two groups were evaluated using an unpaired two-tailed Student's $t$-test. Exact P values and replicate numbers ($n$) are reported in the corresponding figure legends. Data points with significant deviations from the normal range were excluded from the analysis. All attempts at replication were successful, and consistent results were obtained across independent experiments.

## Ethics

We have complied with all relevant ethical regulations for animal use.

## Data availability

The diffraction data and atomic coordinates have been deposited in the PDB under the following accession codes: PTP1B-APO (PDB ID 9LP5) and PTP1B-FU (PDB ID 9LOK). All the data needed to evaluate the conclusions are presented in the paper or the Supplementary Materials. Supplementary data (Supplementary Data 1–4) are provided in this paper.

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

## Acknowledgements

We extend our gratitude to the staff at beamline BL18U1 of the Shanghai Synchrotron Radiation Facility for their valuable assistance.

## Author contributions

J.Z., D.L., and Z.Y. conceived and designed the project; R.S., M.D., W.L., and Y. Liu established and optimized the methodology; J.Z., L.L., N.G., and Yi. Li performed the investigations and data collection; D.L. and Z.Y. provided essential resources and supervision; J.Z., L.L., and Z.Y. prepared the original draft of the manuscript; J.Z., Ma J., K.M., K.W., Z.Y., and Yang G. critically reviewed and edited the manuscript; Project administration was coordinated by Z.Y. and D.L. All the authors reviewed the results and approved the final version of the manuscript.

## Funding

Z.Y. discloses support for the research of this work from the Fundamental and Interdisciplinary Disciplines Breakthrough Plan of the Ministry of Education of China [no.JYB2025XDXM503], the National Natural Science Foundation of China [no.32372580], and the National Key Research and Development Program of China [no.2025YFC3409400]. D.L. discloses support for the research of this work from the Beijing-Tianjin-Hebei Basic Research Cooperation Special Project [no.C2022201099/22JCZXJC00150]. J.M. discloses support for the research of this work from the National Natural Science Foundation of China [no.92156025]. M.D. discloses support for the research of this work from the Tianjin Natural Science Foundation of China [no.23JCQNJC00760].

## Competing interests

The authors declare no competing interests.
