## [Transparent Peer Review file · Communications Biology]

Structural Basis of Fumosorinone-Mediated Allosteric Inhibition of PTP1B for Cancer Immunotherapy

Corresponding Author: Professor Zhiguang Yuchi

Version 0:

Reviewer comments:

Reviewer #1

(Remarks to the Author)

In this manuscript, Jung Zhang et al. examine the antitumor efficacy of fumosorinone (FU) in a mouse model of colon cancer. The authors also report the successful X-ray structure of co-crystallized FU bound to the PTP1B enzyme. While I find the article quite intriguing, there are several areas that could benefit from improvement:

1. The study utilizes recombinant PTP1B; however, it would be helpful to specify the exact amino acid range of the enzyme used, for example, hPTP1B1-285 or hPTP1B1-400.
2. The sodium orthovanadate inhibition assay yielded an IC₅₀ value exceeding 200 micromolar. This result is notably high, as most literature reports an IC₅₀ of sodium orthovanadate of less than 2 micromolar. I recommend repeating this trial.
3. The authors should clarify the rationale behind selecting a dose of 100 mg/kg FU. Additionally, it would be beneficial to discuss the absence of a positive control in the in vivo studies.
4. While the molecular docking and dynamics findings are presented convincingly, a more thorough discussion of these results would enhance the overall understanding of the study.

Reviewer #2

(Remarks to the Author)

The manuscript by Zhang et al. presents a comprehensive and interesting study identifying Fumosorinone (FU) as an allosteric inhibitor of PTP1B with promising implications for cancer immunotherapy. The authors provide a solid multi-disciplinary approach, integrating in vivo tumor models, enzymatic kinetic assays, and high-resolution structural biology to elucidate FU's mechanism of action. The study is well-designed, and the findings are relevant to the field of targeted immunotherapy. However, some minor revisions are needed to strengthen the manuscript.

- 1 For the PTP1B enzyme kinetics measurements mentioned around Line 152, the specific values for K_{cat}, K_m, and K_{cat}/K_m should be provided in a table format.
- 2 The manuscript states that mutagenesis confirmed the importance of specific residues in FU binding, but the results only show the impact of these mutations on PTP1B enzymatic activity without showing the actual binding affinity data.
- 3 The authors discuss a structural superimposition of PTP1B-FU and the apo structure, noting an RMSD of 0.45 Å, to suggest that FU stabilizes an open conformation. However, the corresponding figure for this structural comparison is missing and should be provided.
- 4 The presentation of the ITC and MST titration data could be improved
- 5 Line 176, the claim that FU binds to a "previously uncharacterized allosteric site" is not entirely accurate. This FU binding site partially overlaps the inhibitor-binding site previously reported in PDB 1T49 (Reference 22). The authors should modify this statement to reflect the literature accurately.
- 6 In the Discussion, the authors attempt to correlate the loss of activity in the E276A and N193A mutants with previous reports showing that a K197A substitution or deletion of helix α7 reduces PTP1B activity; the logic of this comparison/correlation is not clearly conveyed.

7 It would be better to use one-letter or three-letter for amino acids throughout, not a mix.

8 Figure S2 lacks panel B and doesn't match S2 figure legend.

Reviewer #3

(Remarks to the Author)

In this study, Zhang et al. demonstrated that fumosorinone, a natural product derived from *Isaria fumosorosea*, serves as a potent and selective inhibitor of PTP1B by binding to a previously uncharacterized allosteric pocket. PTP1B is intimately involved in the development of obesity, type 2 diabetes, and various cancers; therefore, the identification and characterization of its allosteric inhibitors are of significant scientific and medical importance. The authors show that fumosorinone inhibits tumor growth in mice, binds to a distinct allosteric site of PTP1B, stabilizes PTP1B in an inactive conformation, and targets TCPTP as well. These analyses are well organized and reach compelling conclusions. However, before a final decision, I suggest several additional experiments, as listed below.

1. While the authors present data from mouse models and biochemical experiments, I suggest that they also provide data demonstrating whether fumosorinone inhibits PTP1B function at the cellular level, such as by examining the phosphorylation levels of PTP1B substrates.
2. The authors showed that fumosorinone binds to PTP1B with substantial affinity, as determined by ITC and MST. I suggest that they also present the binding affinities of fumosorinone toward mutant forms of PTP1B, such as N193A, F196A, F196R, or W291A.
3. I suggest that the authors present a comparative structural figure showing PTP1B bound to fumosorinone alongside previously reported competitive and noncompetitive PTP1B inhibitors, to clearly illustrate and compare their respective binding sites.

Version 1:

Reviewer comments:

Reviewer #1

(Remarks to the Author)

The authors have appropriately addressed the comments regarding the manuscript. Accepted.

Reviewer #2

(Remarks to the Author)

I have reviewed the revised manuscript and the authors' response to my initial comments. The authors have adequately addressed all of my concerns and have made satisfactory improvements to the paper. I have no further questions or comments.

Reviewer #3

(Remarks to the Author)

I consider that the authors have addressed all the points I indicated, and thus this manuscript is worthy of publication.

Dear Reviewers,

We thank you and all the reviewers for the constructive comments and suggestions, which have led to an overall improvement in our efforts to communicate the results. I have reproduced your comments verbatim below together with our responses. For clarity, our response is shown in dark blue.

Reviewer #1 (Remarks to the Author):

In this manuscript, Jung Zhang et al. examine the antitumor efficacy of fumosorinone (FU) in a mouse model of colon cancer. The authors also report the successful X-ray structure of co-crystallized FU bound to the PTP1B enzyme. While I find the article quite intriguing, there are several areas that could benefit from improvement:

1. The study utilizes recombinant PTP1B; however, it would be helpful to specify the exact amino acid range of the enzyme used, for example, hPTP1B1-285 or hPTP1B1-400.

Authors' Response: We thank the reviewer for this helpful suggestion. The amino acid range of the recombinant human PTP1B used in this study (hPTP1B residues 1-299) was originally described in the Materials and Methods section (Page 12, Line 400). To improve clarity and make this information more immediately accessible to readers, we have now additionally specified it in the Results section as suggested. The revised text now reads: "Recombinant human PTP1B (residues 1-299) was used in all assays." (page 5, Lines 151-152)

2. The sodium orthovanadate inhibition assay yielded an IC₅₀ value exceeding 200 micromolar. This result is notably high, as most literature reports an IC₅₀ of sodium orthovanadate of less than 2 micromolar. I recommend repeating this trial.

Authors' Response: We thank the reviewer for this valuable comment. We agree that sodium orthovanadate is often reported as a potent PTP1B inhibitor, with IC₅₀ values in the low micromolar range under optimized assay conditions. For example, some studies reported IC₅₀ values of ~10 μM or lower (Ha *et al.*, 2020, DOI: 10.1007/s12272-020-01269-4). However, substantially higher IC₅₀ values (100-250 μM) have also been described in truncated constructs or assays containing reducing agents and metal chelators (He *et al.*, 2021, DOI: 10.1016/j.bioorg.2021.104683), conditions that are more comparable to our experimental setup. In our assay, recombinant hPTP1B (residues 1-299) was tested in a buffer containing EDTA, which may affect vanadate speciation and reduce

its apparent inhibitory potency. Importantly, we repeated the sodium orthovanadate inhibition assay multiple times and consistently obtained IC₅₀ values exceeding 200 μM, indicating that this result is reproducible rather than an experimental outlier. In addition, the apparent potency may be influenced by the specific activity and conformational state of the truncated enzyme construct, which could alter vanadate accessibility to the catalytic pocket. Therefore, we believe the observed IC₅₀ (>200 μM) is consistent with values reported under comparable assay conditions and likely reflects the combined effects of assay buffer composition and enzyme construct state.

3. The authors should clarify the rationale behind selecting a dose of 100 mg/kg FU. Additionally, it would be beneficial to discuss the absence of a positive control in the in vivo studies.

Authors' Response: We thank the reviewer for this valuable suggestion. The selection of the FU dose (100 mg/kg) was based on both safety considerations and pharmacokinetic/pharmacodynamic (PK/PD) assessment. Previous animal studies have shown that this dose is well tolerated in rodents without observable toxicity. In addition, our preliminary PK evaluation indicated that FU has an oral bioavailability of approximately 58% and a half-life of ~4.5 h, suggesting that administration at 100 mg/kg can achieve in vivo exposure levels relevant to the inhibitory concentrations observed in our in vitro enzyme assays. We therefore selected this dose as a balance between pharmacological relevance and in vivo tolerability. Regarding the absence of a positive control, we agree that inclusion of a benchmark PTP1B inhibitor would provide an additional point of efficacy comparison. In the present study, however, our primary objective was to establish the in vivo antitumor efficacy and mechanistic immunomodulatory activity of FU as a newly identified PTP1B inhibitor. Because the antitumor efficacy of established PTP1B inhibitors, including AC484, has already been demonstrated in previous in vivo studies (Reference 6), we focused this work on defining whether FU itself can reproduce a therapeutically relevant phenotype. The robust tumor suppression and immune remodeling effects observed in our models provide strong proof-of-concept support for FU activity. We have now added a statement in the Discussion noting that future side-by-side comparisons with benchmark PTP1B inhibitors will be valuable for further positioning the therapeutic potential of FU (page 10, Lines 332-342).

4. While the molecular docking and dynamics findings are presented convincingly, a more thorough discussion of these results would enhance the overall understanding of the study.

Authors' Response: We thank the reviewer for this constructive suggestion. In response, we have substantially expanded the discussion of the molecular docking and molecular dynamics analyses to better clarify how these computational results support the proposed binding mode, conformational stability, and inhibitory mechanism of FU toward PTP1B. The revised text now provides a more explicit interpretation of the key residue interactions and dynamic features observed during the simulations, thereby strengthening the mechanistic connection between the computational and experimental findings. These additions have been incorporated into the revised manuscript (page 9, lines 274-278).

Reviewer #2 (Remarks to the Author):

The manuscript by Zhang et al. presents a comprehensive and interesting study identifying Fumosorinone (FU) as an allosteric inhibitor of PTP1B with promising implications for cancer immunotherapy. The authors provide a solid multi-disciplinary approach, integrating in vivo tumor models, enzymatic kinetic assays, and high-resolution structural biology to elucidate FU's mechanism of action. The study is well-designed, and the findings are relevant to the field of targeted immunotherapy. However, some minor revisions are needed to strengthen the manuscript.

1. For the PTP1B enzyme kinetics measurements mentioned around Line 152, the specific values for K_{cat} , K_m , and K_{cat}/K_m should be provided in a table format.

Authors' Response: We thank the reviewer for this helpful suggestion. In response, we have added a new Table 1 summarizing the specific K_{cat} , K_m , and K_{cat}/K_m values obtained from the PTP1B enzyme kinetics measurements. Presenting these parameters in tabulated form improves the clarity and accessibility of the kinetic data for readers. The new table is now cited in the revised manuscript (page 6, line 157).

2. The manuscript states that mutagenesis confirmed the importance of specific residues in FU binding, but the results only show the impact of these mutations on PTP1B enzymatic activity without showing the actual binding affinity data.

Authors' Response: We thank the reviewer for this insightful comment. In response, we have supplemented the revised manuscript with isothermal titration calorimetry (ITC) measurements for the mutant proteins, performed under the same experimental conditions used for wild-type PTP1B (Figure S5). Under these conditions, no detectable binding signal was observed for the mutants, indicating that the binding affinity was substantially

weakened beyond the reliable detection range of ITC. Nevertheless, our enzymatic inhibition assays suggest that FU retains residual weak interactions with these mutant proteins, as evidenced by its still measurable but markedly reduced inhibitory activity compared with the wild-type enzyme. Together, these results support the conclusion that the mutated residues play critical roles in stabilizing FU binding. We have incorporated the new ITC data and clarified this interpretation in the revised Results section (page 7, lines 192-195).

3. The authors discuss a structural superimposition of PTP1B-FU and the apo structure, noting an RMSD of 0.45 Å, to suggest that FU stabilizes an open conformation. However, the corresponding figure for this structural comparison is missing and should be provided.

Authors' Response: We thank the reviewer for this helpful suggestion. In response, we have now included the structural superimposition of the PTP1B-FU complex with the apo structure in the Supporting Information as Figure S6. The superimposed structures show an overall RMSD of 0.45 Å, providing a direct visual comparison of the conformational relationship between the two states. This figure further supports our conclusion that FU binding is compatible with and stabilizes the open conformation of PTP1B (page 7, lines 215).

4. The presentation of the ITC and MST titration data could be improved.

Authors' Response: We thank the reviewer for this helpful suggestion. In response, we have improved the presentation of both the ITC and MST titration data in the revised manuscript by enhancing the figure resolution, curve visibility, and axis labeling. We have also optimized the layout of the binding panels to make the titration profiles and the derived binding parameters easier to interpret. These revisions improve the clarity and readability of the biophysical data presentation.

5. Line 176, the claim that FU binds to a "previously uncharacterized allosteric site" is not entirely accurate. This FU binding site partially overlaps the inhibitor-binding site previously reported in PDB 1T49 (Reference 22). The authors should modify this statement to reflect the literature accurately.

Authors' Response: We thank the reviewer for this insightful comment and fully agree with this point. The FU-binding site indeed partially overlaps with the previously reported

allosteric inhibitor-binding site observed in the PTP1B-BBR complex (PDB 1T49, Reference 22). Accordingly, we have revised the corresponding statement in the manuscript to more accurately reflect the prior literature (page 6, lines 179-181). The updated wording now clarifies that the FU-binding pocket shares a partially overlapping core region with the known allosteric site, while extending into an adjacent region not previously described, thereby more precisely defining both the relationship to prior structures and the novelty of our finding.

6. In the Discussion, the authors attempt to correlate the loss of activity in the E276A and N193A mutants with previous reports showing that a K197A substitution or deletion of helix $\alpha 7$ reduces PTP1B activity; the logic of this comparison/correlation is not clearly conveyed.

Authors' Response: We thank the reviewer for this valuable comment. In response, we have revised the Discussion to clarify the mechanistic basis for this comparison. Specifically, we now emphasize that N193, K197, and E276 are positioned within the structural network surrounding the WPD loop and helix $\alpha 7$, regions that are critically involved in the catalytic conformational transition of PTP1B. Accordingly, mutations such as E276A or N193A, analogous to the previously reported K197A substitution or helix $\alpha 7$ deletion, are expected to perturb the conformational coupling required for WPD loop closure, thereby reducing enzymatic activity. The revised text now makes this mechanistic relationship more explicit (page 10, lines 307-310).

7. It would be better to use one-letter or three-letter for amino acids throughout, not a mix.

Authors' Response: We thank the reviewer for this careful observation. To improve consistency and readability, we have revised the manuscript and all associated figures to use a uniform one-letter amino acid notation throughout.

8. Figure S2 lacks panel B and doesn't match S2 figure legend.

Authors' Response: We thank the reviewer for identifying this inconsistency. We have now corrected Figure S2 to include panel B and updated the corresponding figure legend to ensure full consistency between the figure and its description.

Reviewer #3 (Remarks to the Author):

In this study, Zhang et al. demonstrated that fumosorinone, a natural product derived from *Isaria fumosorosea*, serves as a potent and selective inhibitor of PTP1B by binding to a previously uncharacterized allosteric pocket. PTP1B is intimately involved in the development of obesity, type 2 diabetes, and various cancers; therefore, the identification and characterization of its allosteric inhibitors are of significant scientific and medical importance. The authors show that fumosorinone inhibits tumor growth in mice, binds to a distinct allosteric site of PTP1B, stabilizes PTP1B in an inactive conformation, and targets TCPTP as well. These analyses are well organized and reach compelling conclusions. However, before a final decision, I suggest several additional experiments, as listed below.

1. While the authors present data from mouse models and biochemical experiments, I suggest that they also provide data demonstrating whether fumosorinone inhibits PTP1B function at the cellular level, such as by examining the phosphorylation levels of PTP1B substrates.

Authors' Response: We thank the reviewer for this valuable suggestion. We fully agree that demonstrating cellular inhibition of PTP1B signaling, for example by examining the phosphorylation levels of established PTP1B substrates in CD8⁺ T cells, would provide important additional mechanistic support for the action of Fumosorinone. In the current study, CD8⁺ T cells isolated from tumor-bearing mice were primarily used for the functional immune analyses already presented in the manuscript. Unfortunately, we did not perform phospho-specific Western blot analyses at the time of sample collection. As a result, addressing this point at the revision stage would require new animal experiments to re-isolate sufficient CD8⁺ T cells, followed by optimization of multiple immunoblot assays using a dedicated panel of phospho-specific antibodies. Given the substantial time, cost, and additional animal use required for this workflow, we believe this set of experiments falls beyond the scope of the current revision. To acknowledge this important point, we have now added a statement in the Discussion noting this limitation and outlining our future plan to systematically examine PTP1B substrate phosphorylation and downstream signaling changes in tumor-infiltrating CD8⁺ T cells (page 10, Lines 322-328).

2. The authors showed that fumosorinone binds to PTP1B with substantial affinity, as determined by ITC and MST. I suggest that they also present the binding affinities of fumosorinone toward mutant forms of PTP1B, such as N193A, F196A, F196R, or W291A.

Authors' Response: We thank the reviewer for this insightful suggestion. In response, we have added ITC measurements for the PTP1B mutants N193A, F196A, and F196R, performed under the same experimental conditions used for the wild-type enzyme (Figure S5). Under these conditions, no detectable binding signal was observed for any of the tested mutants, indicating that the interaction was substantially weakened beyond the reliable detection range of ITC. Consistent with this result, our enzymatic inhibition assays showed that FU retained only markedly reduced inhibitory potency toward these mutants compared with wild-type PTP1B, supporting the conclusion that these residues are critical contributors to FU binding. We have incorporated the new ITC data and the corresponding interpretation into the revised Results section (page 7, lines 192-195).

3. I suggest that the authors present a comparative structural figure showing PTP1B bound to fumosorinone alongside previously reported competitive and noncompetitive PTP1B inhibitors, to clearly illustrate and compare their respective binding sites.

Authors' Response: We thank the reviewer for this valuable suggestion. In response, we have revised the Supplementary Information to include an updated comparative structural annotation in Figure S3, in which the binding position of fumosorinone (FU) is displayed alongside representative competitive inhibitors bound in the catalytic pocket as well as previously reported allosteric/noncompetitive inhibitors. This side-by-side structural comparison more clearly illustrates the spatial relationship among these binding sites and highlights the distinct allosteric binding mode of FU relative to both classical active-site inhibitors and known allosteric ligands. We believe this addition substantially improves the structural context and helps readers more intuitively appreciate the novelty of the FU-binding site.

Best regards,

Zhiguang Yuchi